# Temporal search persistence, certainty, and source preference in dentistry: Results from the National Dental PBRN

**Kimberley R. Isett** [1]⊗*, **Ameet M. Doshi**[2,3]⊗, **Simone Rosenblum**[3‡], **Warren Eller**[4‡], **Diana Hicks**[3‡], **Julia Melkers**[3‡], **the National Dental PBRN Collaborative Group**[¶]

**1** Biden School of Public Policy and Administration, University of Delaware, Newark, Delaware, United States of America, **2** Princeton University Library, Princeton, NJ, United States of America, **3** School of Public Policy, Georgia Institute of Technology, Atlanta, GA, United States of America, **4** John Jay College of Criminal Justice, City University of New York, New York, NY, United States of America

⊗ These authors contributed equally to this work.
‡ WE, DH and JM also contributed equally to this work.
¶ Membership of the National Dental PBRN can be found online at: http://www.nationaldentalpbrn.org/collaborative-group.php.
* kri@udel.edu

**Data Availability Statement:** Data are available from the National Dental PBRN Collaborative website at: https://www.nationaldentalpbrn.org/recruiting-ongoing-upcoming-completed/. The

## Abstract

### Objectives

The primary goal of this paper was to investigate an old question in a new way: what are the search patterns that professionals demonstrate when faced with a specific knowledge gap?

### Methods

We examine data from a cascading survey question design that captures details about searching for information to answer a self-nominated clinical question from 1027 dental professionals enrolled in the National Dental Practice Based Research Network. Descriptive and conditional logistical regression analysis techniques were used.

### Results

61% of professionals in our sample choose informal sources of information, with only about 11% looking to formal peer reviewed evidence. The numbers of professionals turning to general internet searches is more than twice as high as any other information source other than professional colleagues. Dentists with advanced training and specialists are significantly more likely to consult peer-reviewed sources, and women in the sample were more likely than men to continue searching past a first source.

### Conclusions

Speed/availability of information may be just as, or in some cases, more important than credibility for professionals' search behavior. Additionally, our findings suggest that more insights are needed into how various categories of professionals within a profession seek information differently.

study is listed in the section "Conducted between 2012-2020", and is listed under the "Rapid Disruptions" tab.

**Funding:** This study was funded by grants U19-DE-22516 and U19-DE-28717 from the National Institute of Dental and Craniofacial Research (JM), https://www.nidcr.nih.gov/. The funders had no role in study design, data collection and analysis, decision to publish, or preparation of the manuscript.

**Competing interests:** The authors have declared that no competing interests exist.

## Introduction

How professionals search for information has implications across every industry and influences the lives of consumers in myriad ways, from product design to medical treatment. In particular, clinicians are faced with rapid improvements in materials and techniques that could benefit their patients, but must find time to learn about and integrate them into existing practice routines while also continuously seeing patients. There is also an increased volume of publications for clinicians to sort through to find what they need [1]. If this is not enough, new modes of information delivery such as Web 2.0 interfaces provide an even more diffuse set of knowledge, sometimes based on formal peer-reviewed literature, but more often based on practical clinical experience [2]. All of these rapid improvements and the fast availability of information exacerbate the phenomenon of information overload for clinicians.

Even in the face of busy schedules and information overload, there is an expectation that clinicians are up to date on emerging trends. The evidence-based practice paradigm assumes that professionals seek out, are comfortable with, and can identify high quality information sources. Further, patients assume professionals persist in informing themselves about clinical advances in their quest to fill knowledge gaps. A sizeable literature has examined the sources clinicians consult, which documents the "what" of clinician information seeking [3–6].

A characteristic of this source literature is asking individuals to catalogue the sources they typically consult in their professional lives. This method has provided censuses of the clinical journals professionals say they consult on a regular basis [7–9]. While this approach may capture general source preferences and perceived credibility, it treats the search process as relatively static and nonspecific. What is missing from this literature is a more dynamic approach to this question–which sources or source types are preferred as the first or "go to" source, especially when faced with a specific clinical uncertainty? What sources are secondary when the first source doesn't yield actionable information? While qualitative information exists about temporal search within professional health settings [10], the use of quantitative approaches to investigate this phenomenon has been limited [11].

We build on prior work and address the temporal quantitative and specificity gap by introducing a novel approach to better understand search patterns and information outlets among clinical professionals, using dental professionals as a generalizable case. Using a cascading survey question design and then analyzing responses via a conditional logit model, we begin to answer an old question in a new, dynamic way: what are the search patterns that health professionals demonstrate when faced with a specific clinical knowledge gap?

## Background

There exist multiple illustrative models that explain how professionals seek out information [12, 13]. While many of these models are important and influential, such as: "sense-making" [14, 15], Zipf's "Principle of Least Effort" [16], Cyert and March's [17] descriptions of information search processes in organizations, and Tversky and Kahneman's [18] research on individual judgements about information access costs and benefits, we rely upon Robson and Robinson's Information Seeking and Communication Model (ISCM) [19]. The ISCM is a holistic model that convenes characteristics of the information need, sources, search processes, and users to inform a global view of the elements important to information seeking. Most importantly for our purposes, the model incorporates interactive searching based on user assessment of utility and credibility of information, both characteristics that align with an evidence-based practice approach in health professions. ISCM is unique among human information behavior models because it accounts for the interplay of information seeking and sharing

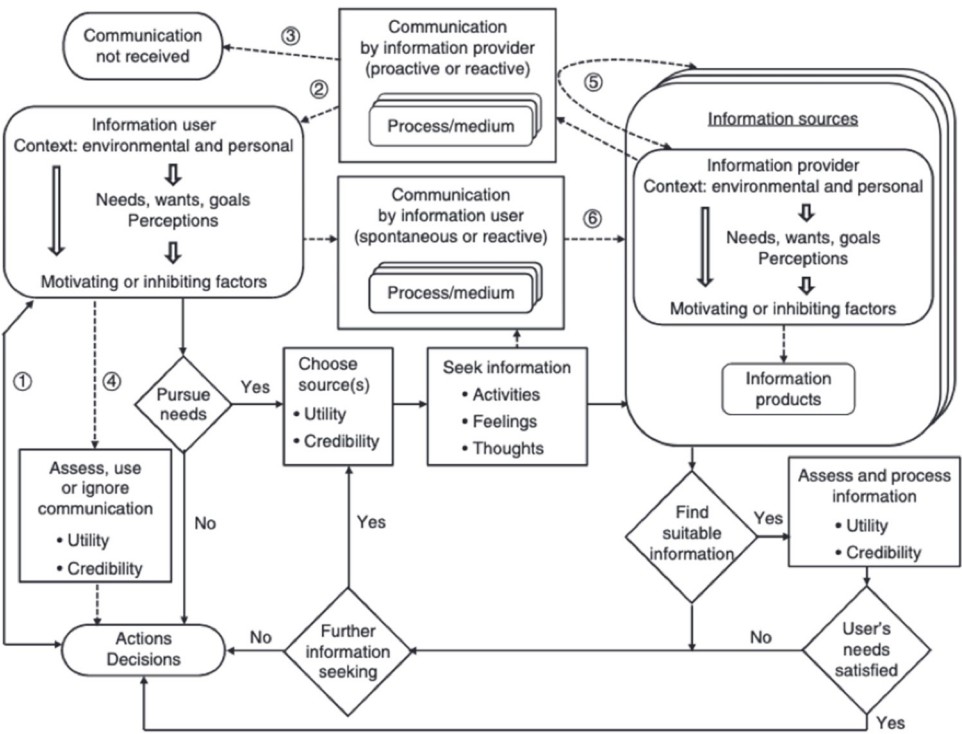

**Fig 1. Information Seeking and Communication Model (ISCM) (Robson and Robinson, 2015).**

among information providers, as well as dynamic assessment of utility and credibility of information (Fig 1).

The ICSM model was developed to describe the cyclical and interactive processes of professional information seeking generally. The model is well suited as a starting point to illustrate the temporality of information behavior as well -a new application of the model that will add some specificity to how we understand iterative search processes. Notably, the effort required to access information sources is a key factor in information behavior studies. For example, Gerstberger and Allen [20] found that accessibility, as a function of both proximity (personal reference books) and usability, was the primary factor determining source selection within a small sample of engineers ($n$ = 19). Their study concluded that "ease of access" and "ease of use" were predictive factors in source selection, suggesting that be the basis for the information behavior of this small population of professionals. The ISCM includes such effort required as "motivating or inhibiting factors" that are inclusive of environmental and personal contexts such as level of training in evidence-based techniques [9, 19]. How motivating and inhibiting factors interact with other factors relevant to information behavior has received less concerted empirical investigation in the literature.

## Clinical professionals and temporal search behavior

A majority of the literature on health information seeking focuses on consumers and patient behavior [21, 22]. Given the importance of informed decision-making by clinicians on patient outcomes, it is remarkable that researchers have not devoted more attention to information behaviors of professionals in health care settings. Keeping up to date with new developments in medicine has become increasingly challenging as the peer-reviewed medical literature has grown in size and scope over the past 25 years [23, 24]. This growth also coincides with

physicians' perceptions of being squeezed for time with patients and colleagues [25], documenting a perceived time famine among clinicians.

A smaller, yet growing, clinically oriented literature investigates dental professional information seeking [26]. Landry et al. [4] drawing from seminal work by Leckie, et al. [13], found that the traditional sources of colleagues and textbooks were primarily consulted when clinical questions arose, consistent with findings from physician-focused studies. And while Botello-Harbaum et al. [7] and Funkhouser et al. [8] illustrated that dental professionals report reading a consistent set of professional journals, recent work by Melkers et al. [2] illustrate that the internet plays an important supplementary role in information diffusion among dental clinicians. The use of online sources to find clinically relevant information has grown substantially over the past two decades [1, 3, 4, 6, 27, 28].

The studies cited above note that there have been changes to the kinds of sources leveraged in information seeking, but research has also noted that information seeking processes also change over time within and across searches [29]. Temporal information behavior is governed by intrinsic factors such as urgency of need and the ability to discern information credibility (e.g. learning), as well as extrinsic factors such as ease of access, time pressure, and the degree to which the source(s) found are acceptable enough (e.g. satisficing) [19, 30, 31]. Adding to the established intrinsic motivators on information search is a newer concept that merits investigation: a "culture of certainty" that can lead professionals to devalue the role of uncertainty in new knowledge creation and create limiters on information search [32].

The work presented here investigates an understudied context within human decision-making settings: the temporal sequencing of clinician decision-making [11]. Given the consistency of professional information seeking patterns already established in the broader literature [13], and among clinical professionals, it is reasonable to expect our conclusions to be generalizable to other clinical and professional settings, and be an early contribution to Web 2.0 contexts [33].

## Methods

### Survey data

Working with our partners at the National Dental Practice Based Research Network (PBRN, "network"), we implemented an online survey of dental clinicians in the United States who are members of the network. The purpose of the survey was to understand information seeking behaviors by dental professionals in order to gain insight into ways we might improve the uptake of evidence-based research into clinical practice. The online survey was conducted using Sawtooth Lighthouse Studio 9.0.1, and the full survey instrument can be accessed through the National Dental PBRN's website [34] or as supplemental materials from the journal. The online survey was approved by the applicable Institutional Review Boards, including the PI's home institution (Georgia Tech) and the Collaborative Network's PI's home institution (University of Alabama at Birmingham). The approvals were provided in the standard format for each Board, typically written. Informed consent was obtained from survey participants prior to the software allowing respondents from entering the survey instrument.

The survey sample was drawn from all 5000 dentists and hygienists enrolled in full participation in the network as of April 2016. The final sample included 3106 clinicians from all six regions. Racial and gender minorities (female dentists and male hygienists) were sampled with certainty. Due to the nature of the research question and our interest in understanding the information behaviors of professional dental clinicians outside of research institutions, we excluded clinicians in dental schools and university dental clinics. Also, because of a broader focus of the project on smoking and use of novel nicotine products (NNPs) we also excluded

orthodontic and pediatric dentists whose client populations have low prevalence of smoking. The survey was launched in August 2016 using an initial email invitation as well as a printed invitation sent by mail. Regional coordinators for the network sent emails and made phone calls to eligible members to encourage participation. Four email reminders were sent at regular monthly intervals by the study team. The survey was closed in December 2016. Participants were given the option to receive a $50 payment card for their participation in the survey.

This paper focuses on a subsample of 1027 respondents (34% of the full 2984 sample and 55% of the 1842 usable responses) who answered a set of questions about searching for information to answer a self-nominated clinical question. This subset includes 81% dentists and 19% hygienists. Clinicians have been found to be selective about which clinical issues they pursue based on their perceptions of the tractability of the problem and whether they consider it to be an "important matter" [35]. Survey respondents were representative of the population of the National Dental PBRN with respect to age and proportion of specialists to generalists, but over represents minorities and gender, both with respect to the network and the profession overall [36].

**Dependent variable.** We asked respondents to explain briefly (70 characters or less) a recent case where they did not have sufficient information to answer a clinical question. We then asked respondents to walk us through their information search in a repeating step-wise manner. Specific to their nominated question they were asked to tell us where they went first to find information, providing the following options:

- I reached out to someone I know

- I consulted professional peer-reviewed published materials

- I consulted other published materials

- I consulted professional sources (dental or other organization)

- I searched on the internet (general internet search)

- I went to a specific website (which one)

- Other (please specify)

After indicating where they went first, we asked whether they stopped here or continued searching. If they stopped, we asked why they had stopped, with the following response choices:

- Source provided enough information to address the problem

- Had to make an immediate decision

- No time to search more

- Decided it was not a problem I could address

- Decided it was not a problem

- Other reasons

If the respondent indicated they continued searching, we asked where they went second, providing the same list of choices. The follow-up question of what they did next was repeated, and at the third level respondents were asked to say what other sources they consulted, checking all that apply.

**Control and independent variables.** We used a variety of variables tin our analyses to ensure that our effects were attributable to the search processes and not to other factors that

could affect search patterns. Our two control variables were certainty ("How certain are you that you could find information on [your self-nominated clinical question] that would help you in your practice?" with four responses on a Likert scale ranging from not at all certain to highly certain), and type of clinical question (the nominated clinical issues were categorized into four broad categories by our clinician team: general dentistry, specialist, unclassifiable, and novel nicotine/smoking cessation). Independent variables fell into three broad categories: individual demographics (gender and race), professional characteristics (professional age (years since earning professional degree), professional title/training (hygienist, dentist, dentist with advanced training, and dental specialist)), and practice characteristics (number of dentists in practice, number of hygienists in practice, whether there is a specialist in the office, and practice location (inner city through rural)).

Table 1 provides the descriptive statistics for the variables used in our models. Within our sample, 46% of our respondents were female and 27% were minorities. We see that on average our respondents have been out of dental or hygiene school for 23 years. Among the 81% of dentists, 37% had advanced training and an additional 10% were dental specialists. In looking at practice characteristics, half of our sample works in a practice with only one dentist, and the majority of our respondents have 1–3 dental hygienists in their practice. More of our sample is located in suburban practices (45%) compared to inner city (12%), urban (27%) and rural (15%) practices.

## Limitations

There are several limitations with our data. First, these data are self-reported, and thus we can only rely on individuals' recollection of their search patterns–though sociologists have shown that individuals can accurately recall "typical" behavior [37]. Additionally, the network population may be different from that of the dentist and hygienist population in the US. However, in clinical assessments the network has been shown to be consistent with dentists at large [38, 39].

## Analysis

Analyses for this paper were performed using Stata v. 14.2 and IBM SPSS Statistics software v. 24. We analyzed the data using correlations, a set of stepped logit models, and a set of stepped conditional logit models to examine the effects of different characteristics on search patterns and information source choices. The use of the stepped logit models allowed us to see whether there were statistically significant effects for some characteristics that were subsumed by additional characteristics in our more complex models.

To see whether there were any effects of the choice of first source used on the second level search, we used a conditional logit model. Conditional logit models are frequently used in epidemiology and allow for the comparison of a treatment group to others in the sample. However, this work represents the first application of the conditional logit method to temporal searching by health professionals. In this study, we use the first source choice as the treatment group. Thus, we can see if there are characteristics specific to particular search patterns that are distinct from others in the sample. We applied the Pearson's chi-squared to test for significance between the first level search and the second level search, and only ran models where there was a statistically significant effect of first source choice on the second source choice.

In addition to the logit models and conditional logit models, we also ran independent sample t-tests between those who stopped searching after the first and second source to see whether there were distinct characteristics of individuals who continued searching compared to those who stopped searching. We further examined the reasons individuals stopped

**Table 1.** Descriptive statistics and frequencies.

| Control Variables | N | Min | Max | Mean | Std. Dev. |
|---|---|---|---|---|---|
| Certainty: How certain are you that you could find information on your [nominated clinical question]? | 1027 | 1 | 4 | 2.96 | 0.98 |
| Categories: | | *Count* | *%* | | |
| *Not certain at all* | | *96* | *9.3* | | |
| *Somewhat certain* | | *228* | *22.2* | | |
| *Generally certain* | | *329* | *32.0* | | |
| *Highly certain* | | *374* | *36.4* | | |
| Clinical Question Topic Categories: | 1025 | | | | |
| *General* | | *509* | *49.7* | | |
| *Specialty* | | *320* | *31.2* | | |
| *Unclassifiable* | | *164* | *16.0* | | |
| *Novel nicotine/smoking cessation* | | *32* | *3.1* | | |
| **Individual Characteristics** | 1027 | | | | |
| Female | | 472 | 46.0 | | |
| Male | | 555 | 54.0 | | |
| Minority | | 279 | 27.2 | | |
| White | | 748 | 72.8 | | |
| **Professional Characteristics** | | **Min** | **Max** | **Mean** | **Std. Dev.** |
| Years Since Earning Professional Degree | 1000 | 1 | 60 | 23.84 | 12.15 |
| Title/training: | 1027 | *Count* | *%* | | |
| *Hygienists* | | *195* | *19.0* | | |
| *Dentists* | | *832* | *81.0* | | |
| *Advanced training* | | *380* | *37.0* | | |
| *Dental Specialist* | | *102* | *9.9* | | |
| **Practice Characteristics** | **N** | **Min** | **Max** | **Mean** | **Std. Dev.** |
| Number of Dentists in Practice | 1015 | 0 | 6 | 2.09 | 1.43 |
| Categories: | | *Count* | *%* | | |
| *0* | | *1* | *0.1* | | |
| *1* | | *507* | *50.0* | | |
| *2* | | *225* | *22.2* | | |
| *3* | | *105* | *10.3* | | |
| *4* | | *66* | *6.5* | | |
| *5–10* | | *83* | *8.2* | | |
| *More than 10* | | *28* | *2.8* | | |
| | **N** | **Min** | **Max** | **Mean** | **Std. Dev.** |
| Number of Hygienists in Practice | 949 | 0 | 6 | 2.49 | 1.54 |
| Categories: | | *Count* | *%* | | |
| *0* | | *72* | *7.6* | | |
| *1* | | *199* | *21.0* | | |
| *2* | | *266* | *28.0* | | |
| *3* | | *190* | *20.0* | | |
| *4* | | *74* | *7.8* | | |
| *5–10* | | *124* | *13.1* | | |
| *More than 10* | | *24* | *2.5* | | |
| Specialist in office | 1027 | 169 | 16.5 | | |
| Practice location | 1020 | | | | |
| *Inner City or Urban Area* | | *123* | *12.1* | | |
| *Urban (Not Inner City)* | | *279* | *27.4* | | |

*(Continued)*

**Table 1.** (Continued)

| Control Variables | N | Min | Max | Mean | Std. Dev. |
|---|---|---|---|---|---|
| *Suburban* | | 461 | 45.2 | | |
| *Rural* | | 157 | 15.4 | | |

searching to see whether there were patterns among the reasons why respondents chose not to search for additional information.

## Results

The focus of the paper is on the temporal search sequences of clinical professionals. As such, Fig 2 illustrates the temporal search sequence data for the 1027 respondents included in this analysis. Of the total sample, 52% were satisfied with the answers they found in their first or second attempts. 30% continued on to a third round of search and the rest gave up after their first (10%) or second (9%) attempts due to constraints or dissatisfaction.

Crosstabs (not shown) reveal that some types of people reported being more successful in finding answers to their questions. Specialists were likely to report finding an answer to their question, with 59% of dental specialists stating the first or second source they consulted provided enough information to answer their question. Similarly, 58% of men stopped searching within two rounds because the sources consulted provided enough information. In addition to these characteristics, question characteristics also played a role in answerability. Questions classified as "general" were more easily answered, 54% of searchers stopped after two rounds

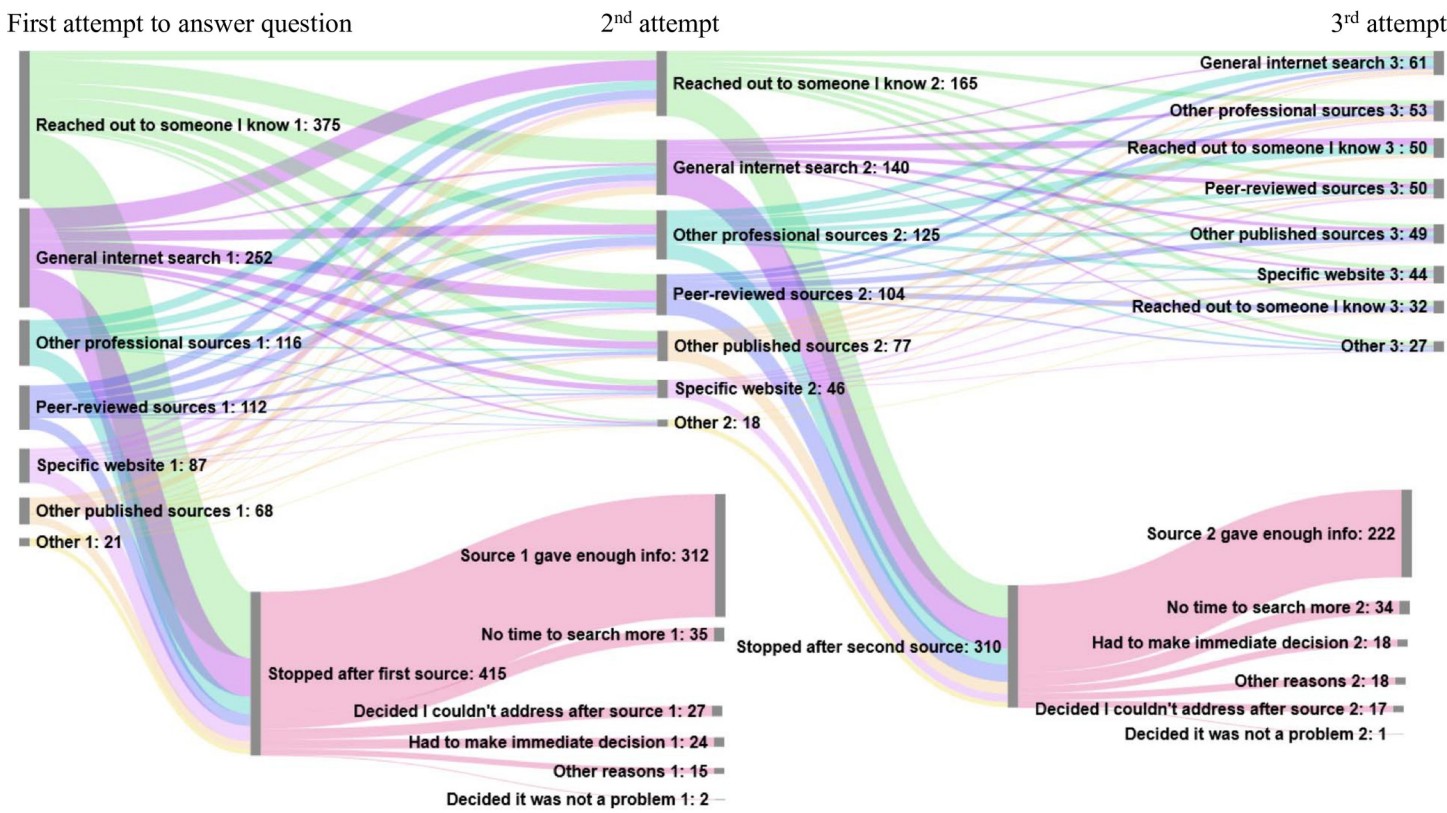

**Fig 2. Three rounds of attempting to answer a clinical question.**

because the sources provided enough information for general questions. Those who were most likely to persist searching were women in the sample, where only 45% stopped in two rounds because the source provided enough information. For questions about novel nicotine or smoking cessation, less than a third stopped searching in the first two rounds.

Some professionals chose to persist in trying to find answers via a different route if the first did not work. When looking at percentages specific to each round of search activity (rather than the overall percentages discussed above), the percentage of searchers who stopped searching because they found a source that provided enough information in the second attempt (72%) was about the same as in the first attempt (76%). The difficulty finding answers and the variety of sources used point to the complexity of questions that arise in practice that are outside the scope of the clinician's current knowledge. When we drill down into the success of each specific strategy, we see that even the most effective strategy (consulting a friend) was only successful about a third of the time. While consults, or asking a friend, were the top choice overall among our sample participants, fewer than half of searchers consulted someone they knew in the first round (36%). Of these, about 34% got the answer they needed. The next most commonly used source in the first round of searching was a general internet search (24%). Of these, 27% stopped searching because the source provided enough information. Those who went to a specific website first (only 8%) had slightly better luck finding information. Of the 8% who used a specific website first, 36% said they stopped searching after the first round because it provided enough information.

Shifting from descriptive to statistical analysis, our results illustrate several characteristics of the information seeker and their practice influenced the choice of first and second round sources within information searches. Table 2 provides the full logit models for each first source used to find information on a respondent's nominated clinical topic. We found that general dentists with advanced training were more than twice as likely as general dentists without advanced training to consult a peer-reviewed source first ($P < .01$), and specialists are more than 3 times as likely than general dentists ($P < .01$). Again, compared to dentists, dental hygienists were less than half as likely to report consulting a professional source first ($P < .05$). We also found that the greater number of dentists there are in a practice, the less likely those individuals are to use professional sources first ($P < .05$). Finally, we saw that each additional year out of dental or hygiene school decreases the likelihood an individual will do a general internet search first ($P < .05$).

The types of problems searched mattered, too. Our results showed that compared to problems classified as general dentistry, those who nominated a topic classified as "specialty" were twice as likely to reach out to someone they know first ($P < .001$) and about half as likely to consult other non-peer-reviewed published material first ($P < .05$) or to do a general internet search first ($P < .01$). In comparison to problems classified as general dentistry, those with problems considered unclassifiable were twice as likely to consult peer-reviewed sources first ($P < .05$).

When we examined the conditional logit models for the second level search patterns (Table 3), we observed that dental specialists were more than four or five times as likely to consult peer-reviewed sources second when their first source was someone they know ($P < .05$), peer-reviewed sources ($P < .01$), or other published material ($P < .01$). We also note that hygienists are half as likely than general dentists to reach out to someone they know as a second option almost regardless of their first source selection (someone they know, other published material, and searched the web (all $P < .05$)).

We saw less impact of topics on second level search patterns. Individuals with specialty topics were about 60% more likely to say they consulted someone they know second after going to someone they know or using other published material, ($P < .05$).

**Table 2. Logit models of first search sources for self-nominated clinical topics (odds ratios reported).**

| VARIABLES | Someone I know | Peer-reviewed | Other published | Professional sources | Searched Web | Specific website | Other |
|---|---|---|---|---|---|---|---|
| **Control Variables** | | | | | | | |
| Clinical question category (general dentistry as reference) | | | | | | | |
| Specialty | 1.923*** | 1.157 | 0.433* | 0.635 | 0.576** | 1.198 | 0.786 |
| Unclassifiable | 1.026 | 2.051* | 0.643 | 1.156 | 0.677 | 0.864 | 1.493 |
| Novel nicotine/smoking cessation | 0.532 | 2.833 | 0.423 | 1.625 | 1.321 | 0.454 | Omitted |
| **Individual characteristics** | | | | | | | |
| Female | 0.790 | 0.981 | 1.150 | 1.201 | 1.135 | 1.120 | 0.936 |
| Minority | 0.968 | 0.845 | 1.107 | 0.932 | 1.110 | 1.285 | 0.402 |
| **Professional Characteristics** | | | | | | | |
| Professional Age | 1.000 | 1.001 | 0.997 | 1.015 | 0.983* | 1.012 | 1.030 |
| Title/training | | | | | | | |
| Hygienist | 1.221 | 1.406 | 0.514 | 0.388* | 1.378 | 0.616 | 2.642 |
| Advanced Training | 0.874 | 2.321** | 0.960 | 1.022 | 0.801 | 0.949 | 1.357 |
| Dental Specialist | 0.873 | 3.759** | 2.388 | 0.578 | 0.626 | 0.692 | Omitted |
| **Practice Characteristics** | | | | | | | |
| Number of Staff Dentists | 0.965 | 1.208 | 0.955 | 0.807* | 1.146 | 0.816 | 0.956 |
| Number of Staff Hygienists | 1.011 | 0.918 | 1.110 | 1.122 | 0.954 | 0.934 | 1.005 |
| Specialist in Office | 1.032 | 0.822 | 0.595 | 1.353 | 0.847 | 1.830 | 1.131 |
| Office location (Inner city urban as reference) | | | | | | | |
| Urban (Not Inner City) | 1.410 | 0.971 | 0.736 | 1.094 | 0.855 | 0.567 | 1.334 |
| Suburban | 1.193 | 0.881 | 0.622 | 0.826 | 1.172 | 1.054 | 0.854 |
| Rural | 1.430 | 0.687 | 1.126 | 0.697 | 1.026 | 0.848 | 0.725 |
| Constant | 0.478* | 0.0505*** | 0.109** | 0.140*** | 0.447* | 0.113*** | 0.00911*** |
| Observations | 913 | 913 | 913 | 913 | 913 | 913 | 807 |

*** p<0.001

** p<0.01

* p<0.05

## Characteristics of "stoppers"

In examining the characteristics of those who stopped searching compared to those who persisted, we saw a few distinct patterns (Table 4). After the first resource consulted, 408 (40%) individuals stopped searching, while 617 continued to search for information. An important finding is that women in our sample, regardless of whether they were a hygienist or dentist, were more likely than men to continue searching for information. We see that in the first round 41% of those who stopped searching after the first source were female, while 49% of those who continued were female ($P < .05$). Additionally, those who were searching for information related to novel nicotine also continued searching ($P < .05$), but no other topics yielded a significant difference. Comparing the difference between means of the certainty variable, certainty also played a role in whether individuals continued or stopped searching in the first round. After the first source, we saw that those who stopped had reported slightly higher levels of certainty that they would find information than those who continued searching ($P < .05$).

After the second resource consulted, those with a topic related to a dental specialty were more likely to continue to a third source, compared to others who consulted a second source. In comparing those who continued past the second source to all who stopped searching at this

**Table 3. Conditional logits of second level search patterns based on first search source for self-nominated clinical topics (odds ratios reported).**

| FIRST | Someone I know | | Peer reviewed | | Other published | | Professional Sources | Searched web |
|---|---|---|---|---|---|---|---|---|
| VARIABLES<br>THEN | Someone I know | Peer reviewed | Peer reviewed | Professional sources | Someone I know | Peer reviewed | Professional sources | Someone I know |
| **Control Variables** | | | | | | | | |
| Clinical question category (general dentistry as control) | | | | | | | | |
| Specialty | 1.654* | 0.862 | 0.951 | 0.693 | 1.605* | 0.808 | 0.700 | 1.528 |
| Unclassifiable | 0.997 | 1.041 | 1.132 | 1.169 | 1.036 | 0.980 | 1.268 | 1.037 |
| Novel nicotine/smoking cessation | 1.277 | 0.494 | 0.582 | 1.762 | 1.531 | 0.420 | 2.098 | 1.259 |
| **Individual Characteristics** | | | | | | | | |
| Female | 1.017 | 0.632 | 0.607 | 0.985 | 0.999 | 0.633 | 0.994 | 0.993 |
| Minority | 1.089 | 1.100 | 1.053 | 0.859 | 1.066 | 1.121 | 0.867 | 1.071 |
| **Professional Characteristics** | | | | | | | | |
| Professional Age | 0.985 | 0.991 | 0.991 | 1.004 | 0.985 | 0.992 | 1.008 | 0.988 |
| Title/training | | | | | | | | |
| Hygienist | 0.464* | 1.393 | 1.557 | 1.237 | 0.461* | 1.435 | 1.136 | 0.421* |
| Advanced Training | 1.128 | 1.503 | 1.703 | 0.927 | 1.109 | 1.541 | 0.972 | 1.167 |
| Dental Specialist | 1.675 | 4.088* | 5.044** | 0.146* | 1.514 | 4.655** | 0.162* | 1.891 |
| **Practice Characteristics** | | | | | | | | |
| Number of Staff Dentists | 0.950 | 1.149 | 1.177 | 0.824 | 0.949 | 1.138 | 0.827 | 0.925 |
| Number of Staff Hygienists | 1.072 | 1.019 | 1.013 | 1.013 | 1.063 | 1.033 | 1.023 | 1.098 |
| Specialist in Office | 1.002 | 0.344* | 0.336* | 1.658 | 0.940 | 0.351* | 1.732 | 0.995 |
| Office location (Inner city urban as control) | | | | | | | | |
| Urban (Not Inner City) | 2.486* | 0.975 | 1.009 | 0.485 | 2.178 | 1.001 | 0.461 | 2.173 |
| Suburban | 2.195 | 1.106 | 1.109 | 0.413* | 1.953 | 1.142 | 0.397* | 1.890 |
| Rural | 1.929 | 0.794 | 0.800 | 0.829 | 1.517 | 0.928 | 0.727 | 1.698 |
| Observations | 546 | 546 | 546 | 546 | 546 | 546 | 546 | 546 |

*** p<0.001

** p<0.01

* p<0.05

point, we see women were more likely to have continued beyond two sources in their search. However, while those who continued past the second source were still slightly less certain than those who stopped, the difference in means between the two groups was not statistically significant.

Table 5 presents crosstabs that illustrates the reasons why people stopped searching. Overwhelmingly, these professionals reported that they ended their search because they were able to find the information they needed, rather than any other reason, regardless of any personal, professional, or topic characteristics. With that said, there were several characteristics that had noticeable differences between relevant comparators. A greater percentage of participants who were white (54%) and those who were male (54%) said that they stopped searching because they believed their source provided enough information. This was true for those who stopped after the first source or second source. Similarly, a larger percentage of dentists (54%), and to an even greater extent dental specialists (59%) said they stopped searching because their source provided enough information at both levels. Compared to dentists, a greater percentage of hygienists said they stopped after the first or second source because they did not have time to

**Table 4. Comparison of individuals who continued searching to those who stopped.**

| | All | Action after first source | | | Action after second source (compared to those who continued past first source) | | | Action after second source (compared to all searchers) | | |
|---|---|---|---|---|---|---|---|---|---|---|
| | | Stopped | Continued | | Stopped | Continued | | Stopped | Continued | |
| | (n = 1027) | (n = 408) | (n = 617) | P-value | (n = 309) | (n = 307) | P-value | (n = 720) | (n = 307) | P-value |
| CONTROL VARIABLES | | | | | | | | | | |
| *Information search certainty* | | | | | | | | | | |
| Certainty ((Mean ± SD) Range = 1–4) | 2.96±0.98 | 3.04±1.00 | 2.90±0.96 | 0.019 | 2.88±0.97 | 2.92±0.95 | 0.621 | 2.97±0.99 | 2.92±.951 | 0.434 |
| *Clinical question category* | % | % | % | | % | % | | % | % | |
| General dentistry topic | 49.7 | 50.4 | 49.4 | 0.750 | 52.9 | 45.6 | 0.070 | 51.4 | 45.6 | 0.090 |
| Dental specialty topic | 31.2 | 31.9 | 30.5 | 0.631 | 26.6 | 34.5 | 0.033 | 29.7 | 34.5 | 0.135 |
| Unclassifiable topic | 16.0 | 16.0 | 16.1 | 0.966 | 16.9 | 15.3 | 0.596 | 16.4 | 15.3 | 0.667 |
| Novel nicotine topic | 3.1 | 1.7 | 4.1 | 0.023 | 3.6 | 4.6 | 0.535 | 2.5 | 4.6 | 0.121 |
| INDIVIDUAL CHARACTERISTICS | | | | | | | | | | |
| Female | 46.0 | 41.0 | 49.0 | 0.012 | 45.0 | 53.0 | 0.053 | 43.0 | 53.0 | 0.003 |
| Minority | 27.0 | 28.0 | 27.0 | 0.612 | 26.0 | 28.0 | 0.615 | 27.0 | 28.0 | 0.859 |
| PROFESSIONAL CHARACTERISTICS | | | | | | | | | | |
| Dentists | 81.0 | 81.9 | 80.4 | 0.557 | 81.9 | 78.8 | 0.342 | 81.9 | 78.8 | 0.270 |
| Hygienists | 19.0 | 18.1 | 19.6 | 0.557 | 18.1 | 21.2 | 0.342 | 18.1 | 21.2 | 0.270 |
| Advanced Training | 36.9 | 37.0 | 36.8 | 0.943 | 35.3 | 38.4 | 0.417 | 36.3 | 38.4 | 0.514 |
| Specialist | 10.0 | 10.6 | 9.7 | 0.662 | 9.4 | 10.1 | 0.766 | 10.0 | 10.1 | 0.957 |
| PRACTICE CHARACTERISTICS | | | | | | | | | | |
| Specialist in Office | 16.5 | 16.4 | 16.7 | 0.909 | 15.9 | 17.6 | 0.565 | 16.2 | 17.6 | 0.586 |
| Inner city/urban office | 12.2 | 13.1 | 11.6 | 0.473 | 10.4 | 12.9 | 0.332 | 12.0 | 12.9 | 0.685 |
| Urban (not inner city) office | 27.4 | 26.2 | 28.2 | 0.474 | 27.2 | 29.0 | 0.610 | 26.6 | 29.0 | 0.417 |
| Suburban office | 45.2 | 44.7 | 45.4 | 0.836 | 47.3 | 43.6 | 0.361 | 45.8 | 43.6 | 0.520 |
| Rural office | 15.3 | 16.1 | 14.9 | 0.602 | 15.2 | 14.5 | 0.811 | 15.7 | 14.5 | 0.629 |

search further (11%, compared to only 6% of dentists), and a greater percentage of hygienists than dentists also said they stopped because they decided the issue was not a problem they could address (8%, compared to only 3% of dentists). Finally, when we look at the topics, a greater percentage of those whose questions were unclassifiable stopped searching after the first resource because they had no time.

## Certainty

Finally, we looked at the levels of certainty and why individuals stopped (Table 6). 91% of those who said they were highly certain they would find information and stopped after the first source said they did so because the first source provided enough information, compared to only 33% of those who were not at all certain and stopped after the first source. This held true for the second source, too, with 89% and 27% respectively. While those with higher levels of certainty reported stopping because they found a source with enough information, those with lower levels of certainty reported stopping due to lack of time (either they had to make an immediate decision or had no time to search more), or because they decided it was not a problem they could address.

## Discussion

The primary goal of this paper was to expand an understanding of the ISCM model to empirically address the iterative processes described in that model with novel quantitative analysis.

**Table 5. Reasons for stopping search by sample characteristics.**

| | | | | | | | | | | | | | |
|---|---|---|---|---|---|---|---|---|---|---|---|---|---|
| | | **Reasons for stopping after first source** | | | | | | | | | | | |
| | | **Individual and professional characteristics** | | | | | | | | **Question topic** | | | |
| | All | Male | Female | White | Minority | Hygienists | Dentists | Advanced Training | Specialist | General | Specialty | Unclassifiable | NNP |
| n | 408 | 240 | 168 | 293 | 115 | 74 | 334 | 151 | 43 | 205 | 130 | 65 | 7 |
| | % | % | % | % | % | % | % | % | % | % | % | % | % |
| Enough information | 75.7 | 79.6 | 70.2 | 79.2 | 67.0 | 63.5 | 78.4 | 77.5 | 81.4 | 77.6 | 75.4 | 70.8 | 71.4 |
| Make immediate decision | 5.9 | 6.7 | 4.8 | 6.1 | 5.2 | 2.7 | 6.6 | 6.0 | 7.0 | 5.4 | 4.6 | 9.2 | 14.3 |
| No time | 8.3 | 5.8 | 11.9 | 7.5 | 10.4 | 17.6 | 6.3 | 7.9 | 7.0 | 9.3 | 5.4 | 12.3 | 0.0 |
| Not a problem I could address | 6.4 | 4.6 | 8.9 | 4.4 | 11.3 | 12.2 | 5.1 | 4.6 | 2.3 | 5.9 | 7.7 | 4.6 | 14.3 |
| Decided it was not a problem | 0.5 | 0.4 | 0.6 | 0.3 | 0.9 | 0.0 | 0.6 | 0.7 | 0.0 | 0.0 | 0.8 | 1.5 | 0.0 |
| Other reasons | 3.2 | 2.9 | 3.6 | 2.4 | 5.2 | 4.1 | 3.0 | 3.3 | 2.3 | 2.0 | 6.2 | 1.5 | 0.0 |
| | | **Reasons for stopping after second source** | | | | | | | | | | | |
| | | **Individual and professional characteristics** | | | | | | | | **Question topic** | | | |
| | All | Male | Female | White | Minority | Hygienists | Dentists | Advanced Training | Specialist | General | Specialty | Unclassifiable | NNP |
| n | 310 | 170 | 140 | 230 | 80 | 56 | 254 | 109 | 29 | 164 | 82 | 52 | 11 |
| | % | % | % | % | % | % | % | % | % | % | % | % | % |
| Enough information | 71.6 | 75.3 | 67.1 | 74.3 | 63.7 | 58.9 | 74.4 | 73.4 | 86.2 | 70.1 | 73.2 | 78.8 | 45.5 |
| Make immediate decision | 5.8 | 4.7 | 7.1 | 5.7 | 6.3 | 7.1 | 5.5 | 3.7 | 3.4 | 5.5 | 4.9 | 9.6 | 0.0 |
| No time | 11.0 | 9.4 | 12.9 | 9.1 | 16.3 | 16.1 | 9.8 | 10.1 | 3.4 | 13.4 | 7.3 | 5.8 | 27.3 |
| Not a problem I could address | 5.5 | 5.9 | 5.0 | 5.2 | 6.3 | 10.7 | 4.3 | 4.6 | 0.0 | 6.1 | 3.7 | 1.9 | 27.3 |
| Decided it was not a problem | 0.3 | 0.6 | 0.0 | 0.4 | 0.0 | 0.0 | 0.4 | 0.9 | 0.0 | 0.0 | 1.2 | 0.0 | 0.0 |
| Other reasons | 5.8 | 4.1 | 7.9 | 5.2 | 7.5 | 7.1 | 5.5 | 7.3 | 6.9 | 4.9 | 9.8 | 3.8 | 0.0 |

We accomplished this goal by analyzing dental professional's search patterns under conditions of clinical uncertainty and investigated how temporal aspects of information search interacted with individual and contextual factors related to the searchers of information. Dental professionals have exhibited patterns consistent with other clinical and professional knowledge-based groups in past studies, thereby making then a reasonable population to study to understand these kinds of patterns for professional knowledge workers generally [4, 13]. We draw out the implications of those patterns here, rounding out the observations with conclusions for the ISCM model.

While there are many possible pathways a professional can take when faced with uncertainty, almost 61% (Fig 2) choose informal sources of information such as reaching out to individuals and a general internet search, with only about 11% looking to formal peer reviewed evidence. These two sources have high accessibility in that they are fast to access and their ease of use is high [12]. Colleagues may have the additional benefit of having high credibility based on long term relationships and practice history. What is interesting here is that the numbers of professionals turning to general internet searches is more than twice as high as any other information source other than professional colleagues.

The use of the internet may speak to time pressure dynamics in clinical settings. The ubiquity of computers in offices and internet access on handheld devices has undoubtedly reduced the effort required to quickly access information from the internet. The availability of the internet, combined with documented perceived time famines for clinicians and hygienists in

**Table 6. Reasons for stopping by respondent certainty of finding information.**

| | Reasons for stopping after first source | | | | |
|---|---|---|---|---|---|
| | | Respondent's level of certainty that s/he will find information about nominated question | | | |
| | All | Not certain at all | Somewhat certain | Generally certain | Highly certain |
| n | 407 | 39 | 79 | 114 | 175 |
| | % | % | % | % | % |
| Source provided enough information | 75.7 | 33.3 | 54.4 | 81.6 | 90.9 |
| Had to make an immediate decision | 5.9 | 7.7 | 8.9 | 7.0 | 3.4 |
| No time to search more | 8.4 | 28.2 | 17.7 | 7.0 | 0.6 |
| Decided it was not a problem I could address | 6.4 | 23.1 | 15.2 | 2.6 | 1.1 |
| Decided it was not a problem | 0.5 | 0.0 | 0.0 | 0.0 | 1.1 |
| Other reasons | 3.2 | 7.7 | 3.8 | 1.8 | 2.9 |
| | Reasons for stopping after second source | | | | |
| | | Respondent's level of certainty that s/he will find information about nominated question | | | |
| | All | Not certain at all | Somewhat certain | Generally certain | Highly certain |
| n | 310 | 30 | 77 | 104 | 99 |
| | % | % | % | % | % |
| Source provided enough information | 71.6 | 26.7 | 61.0 | 76.0 | 88.9 |
| Had to make an immediate decision | 5.8 | 6.7 | 11.7 | 5.8 | 1.0 |
| No time to search more | 11.0 | 20.0 | 19.5 | 8.7 | 4.0 |
| Decided it was not a problem I could address | 5.5 | 16.7 | 5.2 | 5.8 | 2.0 |
| Decided it was not a problem | 0.3 | 3.3 | 0.0 | 0.0 | 0.0 |
| Other reasons | 5.8 | 26.7 | 2.6 | 3.8 | 4.0 |

our sample can lead them to opt for a quick internet search rather than sources they deem more credible. Rosenblum et al [27] documented the tension some clinicians feel with this trade-off, making the rates at which clinicians turn to these types of sources in both their first and second rounds of searching documented here a compelling finding. Perlow's [40] seminal research on time famine and the sociology of how teams operate under time pressure could guide additional research on this question.

Unpacking the aggregate results, we found that important differences existed among discrete types of professionals in our sample. For example, we find that dentists with advanced training and specialists are significantly more likely to consult peer-reviewed sources, and continue to do so at higher rates than other types of professionals in all search rounds. These results suggest that these types of professionals place value on the credibility signaled by the peer-reviewed literature as described by Robson and Robinson's ISCM, differentiating their patterns from the aggregate findings where quickness and rapidity appear to be the driver for selection. Additionally, these data suggest a gender effect with respect to search persistence where women in the sample were more likely than men to continue searching after a first source did not provide an answer -though these results beg the question of whether certain professionals are better at searching, better trained at searching, or just more confident with the information they find. Certainly, the differences found regarding gender identity merit further investigation, and have corollaries in other professional domains [41, 42].

Despite women's persistence in searching, taking more time to resolve uncertainty, the culture of certainty described by von Bergmann and Shuler [32] appears to be alive and well. The data illustrate that the number of conducted searches after the second level declines dramatically. Dental professionals presented with a novel clinical challenge engage in searching one information source, with some persisting to a second source, but rarely continuing on to a

third. Thereby, supporting the notion that these professionals value rapid closure to clinical questions. Investigation into how to train professionals on how to do effective searches or how to make high quality sources more readily available may be valuable in reducing the science to service gap in clinical practice. This finding also suggests that the empirical value of this kind of temporal analysis may be limited to two levels (e.g. a first source, followed by a second) when working in knowledge professions that are prone to cultures of certainty.

The findings in this study provide some illumination for the ISCM model, teasing out some of the underlying dynamics embedded within it. We expand on the ISCM by drawing from Case and Given [12] to suggest the while credibility of the information source matters, that it may not be the only characteristic of the information that matters. In addition to credibility, speed/availability may be just as, or in some cases, more important than credibility. Our findings also suggest that the ISCM model could be further explored to look at questions of "for whom?". A more nuanced understanding of the ISCM could probe the characteristics of the professionals who are seeking information and under what conditions those characteristics matter. Our finding that women persist in search behavior at greater rates than men, as well as that more highly trained professional (in our sample advanced training dentists and specialists) appear to have a narrower set of decision criteria than the aggregate sample are important nuances to the ISCM.

This research fills a significant gap in the literature on how professionals search under conditions of uncertainty, located at the intersection of users and information. Understanding how, and whether, health professionals persist in searching for information to reduce uncertainty is a consequential puzzle worth solving. While our findings may have initiated more questions than we answered, our novel empirical work provides new breadth and nuance to the deep models that currently serve as the cornerstones of what we know about information behavior.

## Acknowledgments

We thank Michael Melkers for contributions to previous versions of this paper; Tracy Shea, Sharon Nicholson Harrell, Sonia Makhija, Jason McCargar, and Richa Singhania who provided initial insight into categories of our dental topics, and the clinicians in the National Dental PBRN who took the time to respond to our survey. An Internet site devoted to details about the nation's network is located at http://NationalDentalPBRN.org. We are also grateful to our entire study team for their insight and contribution to this project (Eugenio Beltran, DMD, MPH, MS, DrPH, DABDPH, George Ford, DMD, Julie Frantsve-Hawley, RDH, PhD, Ellen Funkhouser, DrPH and Dan Meyer, DDS), and the Regional Coordinators who were indispensable in the data collection and follow up process (Meredith Buchberg, MPH, Claudia Carcelén, MPH, Colleen Dolan, MPH, Stephanie Hodge, MA, Hanna Knopf, BA, Shermetria Massingale, MPH, CHES, Deborah McEdward, RDH, BS, CCRP, Christine O'Brien, RDH, Stephanie Reyes, BA, Tracy Shea, RDH, BSDH, and Ellen Sowell, BA). The National Dental PBRN is a consortium of over 5,000 dentists across the United States and led by Gregg H. Gilbert (ghg@uab.edu).

## Author Contributions

**Conceptualization:** Kimberley R. Isett, Diana Hicks, Julia Melkers.

**Data curation:** Simone Rosenblum, Julia Melkers.

**Formal analysis:** Simone Rosenblum.

**Funding acquisition:** Julia Melkers.

**Investigation:** Kimberley R. Isett, Simone Rosenblum, Julia Melkers.

**Methodology:** Kimberley R. Isett, Simone Rosenblum, Warren Eller.

**Project administration:** Kimberley R. Isett, Simone Rosenblum, Diana Hicks, Julia Melkers.

**Supervision:** Kimberley R. Isett, Diana Hicks, Julia Melkers.

**Visualization:** Simone Rosenblum.

**Writing – original draft:** Kimberley R. Isett, Ameet M. Doshi, Simone Rosenblum, Warren Eller.

**Writing – review & editing:** Kimberley R. Isett, Ameet M. Doshi, Simone Rosenblum, Warren Eller, Diana Hicks, Julia Melkers.

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
