## [Decision Letter · Decision Letter 0]

30 Nov 2021

PONE-D-20-28405Temporal Search Persistence, Certainty, and Source Preference in Dentistry:Results from the National Dental PBRNPLOS ONE

Dear Dr. Isett,

Thank you for submitting your manuscript to PLOS ONE. After careful consideration, we feel that it has merit but does not fully meet PLOS ONE’s publication criteria as it currently stands. Therefore, we invite you to submit a revised version of the manuscript that addresses the points raised during the review process.

Three reviewers have evaluated your submission, and have identified several aspects of the manuscript that will need to be carefully revised in order to meet PLOS ONE's publication criteria. Please respond to all of the items raised by the reviewers, paying particular attention to Reviewer 1's comments regarding the methods and interpretation of the results.

We look forward to receiving your revised manuscript.

Kind regards,

Jamie Males

Staff Editor

PLOS ONE

2. a) Thank you for including your ethics statement: "The online survey was approved by the eight applicable Institutional Review Boards (six regions for the study, the PI's home institution, and the Collaborative PI's home institution. The approvals were provided in the standard format for each Board, typically written."

b) Please provide additional details regarding participant consent. In the ethics statement in the Methods and online submission information, please ensure that you have specified (1) whether consent was informed and (2) what type you obtained (for instance, written or verbal, and if verbal, how it was documented and witnessed). If your study included minors, state whether you obtained consent from parents or guardians. If the need for consent was waived by the ethics committee, please include this information.

“This study was funded by grants U19-DE-22516 and U19-DE-28717 from the National Institute of Dental and Craniofacial Research.   We thank Michael Melkers for contributions to previous versions of this paper; Tracy Shea, Sharon Nicholson Harrell, Sonia Makhija, Jason McCargar, and Richa Singhania who provided initial insight into categories of our dental topics,  and the clinicians in the National Dental PBRN who took the time to respond to our survey. An Internet site devoted to details about the nation’s network is located at http://NationalDentalPBRN.org.  We are also grateful to our entire study team for their insight and contribution to this project (Eugenio Beltran, DMD, MPH, MS, DrPH, DABDPH, George Ford, DMD, Julie Frantsve-Hawley, RDH, PhD, Ellen Funkhouser, DrPH and Dan Meyer, DDS), and the Regional Coordinators who were indispensable in the data collection and follow up process (Meredith Buchberg, MPH, Claudia Carcelén, MPH, Colleen Dolan, MPH, Stephanie Hodge, MA, Hanna Knopf, BA, Shermetria Massingale, MPH, CHES, Deborah McEdward, RDH, BS, CCRP, Christine O’Brien, RDH, Stephanie Reyes, BA, Tracy Shea, RDH, BSDH, and Ellen Sowell, BA).”

“This study was funded by grants U19-DE-22516 and U19-DE-28717 from the National Institute of Dental and Craniofacial Research (JM), https://www.nidcr.nih.gov/. The funders had no role in study design, data collection and analysis, decision to publish, or preparation of the manuscript. “

4. One of the noted authors is a group or consortium National Dental PBRN Collaborative Group. In addition to naming the author group, please list the individual authors and affiliations within this group in the acknowledgments section of your manuscript. Please also indicate clearly a lead author for this group along with a contact email address.

Reviewers' comments:

Reviewer's Responses to Questions

**Comments to the Author**

1. Is the manuscript technically sound, and do the data support the conclusions?

Reviewer #1: Yes

Reviewer #2: Yes

Reviewer #3: Yes

2. Has the statistical analysis been performed appropriately and rigorously? 

Reviewer #1: I Don't Know

Reviewer #2: Yes

Reviewer #3: Yes

3. Have the authors made all data underlying the findings in their manuscript fully available?

Reviewer #1: Yes

Reviewer #2: Yes

Reviewer #3: Yes

4. Is the manuscript presented in an intelligible fashion and written in standard English?

Reviewer #1: Yes

Reviewer #2: Yes

Reviewer #3: Yes

5. Review Comments to the Author

Reviewer #1: 1. Please provide data supporting statement of survey being representative of network at-large. Also, it would be helpful to extend this consideration to the practitioner population at the national level.

2. It would be helpful to gain a broader context for this information by providing temporal scale and frequency of practitioner engagement in search behaviors.

3. Clarify discriminating factor that led to this subset of respondents considered for this study from the larger pool of survey completors.

4. Results discussing 52-59% stopping search due to satisfaction need to be expanded to also include levels of dissatisfaction as well.

5. The distinction between these 2 statements need to be clarified: "...52% were satisfied with the answers they found in their first or second attempts." and "The percentage of searchers who stopped searching because they found a source that provided enough information in the second attempt (72%) was about the same as in the first attempt (76%)."

6. Justify this results statement including the definition of 'worth' being applied: "It was worth trying to find answers via a different route if the first did not work."

7. Again, clarification of the findings is needed similar to item 5 above with the following statement: "Of these, about 34% got the answer they needed. The next most commonly used source in the first round of searching was a general internet search (24%). Of these, 27% stopped searching because the source provided enough information. Those who went to a specific website first (only 8%) had slightly better luck finding information. Of the 8% who used a specific website first, 36% said they stopped searching after the first round because it provided enough information." If each search strategy is reaching less than 40% success, how is overall 76% success found for first search?

8. Clarify: "general dentists with advanced training were more than twice as likely as general dentists without advanced training to consult a peer-reviewed source first" when these groups' responses were 11% v 13%.

9. Sex differences in search stoppage are noted. It should be examined and clarified as to how this may reflect a greater persistence or a lesser effectiveness of searching.

10. As the survey was initiated relative to nicotine cessation and alternative tobacco products, there should be greater presentation of how this specific question was addressed relative to the general behaviors reported.

11. Figure 2. Remove the number 1 or 2 after reason descriptors as they are distracting to frequency data

Reviewer #2: This study tackled a topic that has not been previously explored in the depth and breadth described. This study will encourage more researchers to investigate traits in data retrieval amongst oral health providers to achieve best practice, and more importantly how to address misinformation, and encourage seeking peer reviewed resources.

Reviewer #3: Great job on drafting an interesting manuscript that addresses an important research question that is warranted in today’s face-paced clinical environment. The manuscript aims to understand the avenues that clinicians take when faced with scenario where they lack information. The authors conducted a survey among more than 3100 clinicians (dentists, hygienists and specialists) to understand the methods clinicians adopt to retrieve information and address uncertainties. Below are my comments:

Background:

Line 102 – I think the sentence needs rewording.

Methods:

Line 170 – Why did you choose members of the National Dental Practice Based Research Network as your sampling frame? I know you mentioned later that they were not very different than other clinicians, but I think your audience might need to see a rationale for your choice.

Line 174 – Could you please add the survey instrument as an appendix for the convenience of your audience?

Line 201 – Dependent variable: What was the rationale behind asking the participants to provide their own case? Was there any way you could have adjusted for the level of complexity of the case the participant came up with? How different do you think that could have been (dentists vs hygienists) (specialists vs GPs)? Do you think it would have been better to have a more homogenous sample (only dentists, only hygienists or only specialists)?

Discussion:

I think it’s important to elaborate on the implications of your findings re: clinicians prioritizing time over the credibility of resources from the professional and educational perspectives. What should the dental and dental hygiene associations do to make information more readily available for clinicians? How about integrating some of the researching techniques into school curricula? Do you feel clinicians are equipped with the skills that make their searching strategies efficient?

6. PLOS authors have the option to publish the peer review history of their article (what does this mean?). If published, this will include your full peer review and any attached files.

Reviewer #1: No

Reviewer #2: **Yes: **Zaid H Baqain

Reviewer #3: No

---

## [Author Response · Author response to Decision Letter 0]

8 Feb 2022

We have, to the best of our ability, addressed the style formatting included in the style templates.

2. a) Thank you for including your ethics statement: "The online survey was approved by the eight applicable Institutional Review Boards (six regions for the study, the PI's home institution, and the Collaborative PI's home institution. The approvals were provided in the standard format for each Board, typically written."

This statement has been added in the first paragraph in the Methods section

We have added the relevant details in the manuscript.

b) Please provide additional details regarding participant consent. In the ethics statement in the Methods and online submission information, please ensure that you have specified (1) whether consent was informed and (2) what type you obtained (for instance, written or verbal, and if verbal, how it was documented and witnessed). If your study included minors, state whether you obtained consent from parents or guardians. If the need for consent was waived by the ethics committee, please include this information.

We have added this information in the first paragraph of the Methods section

n/a

“This study was funded by grants U19-DE-22516 and U19-DE-28717 from the National Institute of Dental and Craniofacial Research. We thank Michael Melkers for contributions to previous versions of this paper; Tracy Shea, Sharon Nicholson Harrell, Sonia Makhija, Jason McCargar, and Richa Singhania who provided initial insight into categories of our dental topics, and the clinicians in the National Dental PBRN who took the time to respond to our survey. An Internet site devoted to details about the nation’s network is located at http://NationalDentalPBRN.org. We are also grateful to our entire study team for their insight and contribution to this project (Eugenio Beltran, DMD, MPH, MS, DrPH, DABDPH, George Ford, DMD, Julie Frantsve-Hawley, RDH, PhD, Ellen Funkhouser, DrPH and Dan Meyer, DDS), and the Regional Coordinators who were indispensable in the data collection and follow up process (Meredith Buchberg, MPH, Claudia Carcelén, MPH, Colleen Dolan, MPH, Stephanie Hodge, MA, Hanna Knopf, BA, Shermetria Massingale, MPH, CHES, Deborah McEdward, RDH, BS, CCRP, Christine O’Brien, RDH, Stephanie Reyes, BA, Tracy Shea, RDH, BSDH, and Ellen Sowell, BA).”

“This study was funded by grants U19-DE-22516 and U19-DE-28717 from the National Institute of Dental and Craniofacial Research (JM), https://www.nidcr.nih.gov/. The funders had no role in study design, data collection and analysis, decision to publish, or preparation of the manuscript. “

The funding related information has been removed from the text. There are no changes necessary for the funding statement.

4. One of the noted authors is a group or consortium National Dental PBRN Collaborative Group. In addition to naming the author group, please list the individual authors and affiliations within this group in the acknowledgments section of your manuscript. Please also indicate clearly a lead author for this group along with a contact email address.

Added as requested to the acknowledgements and identified appropriately per your styleguide

As noted above, this information has been added in the first paragraph of the methods section.

Comments to the Author

Reviewer #1: 1. Please provide data supporting statement of survey being representative of network at-large. Also, it would be helpful to extend this consideration to the practitioner population at the national level.

This information has been added on page 8 and a reference has been added to another paper where that data is analyzed. And reads:

Survey respondents were representative of the population of the National Dental PBRN with respect to age and proportion of specialists to generalists, but over represents minorities and gender, both with respect to the network and the profession overall (35).

2. It would be helpful to gain a broader context for this information by providing temporal scale and frequency of practitioner engagement in search behaviors.

While the literature has not provided a concrete estimate of how often and how long clinical practitioners engage in search behavior, it does provide some parameters about clinical searches. We have added some language and a reference around that point. It reads:

Clinicians have been found to be selective about which clinical issues they pursue based on their perceptions of the tractability of the problem and whether they consider it to be an “important matter” (35). 

3. Clarify discriminating factor that led to this subset of respondents considered for this study from the larger pool of survey completors.

Although this information was included in the original manuscript, this comment made us realize that the information was somewhat obscured by the way the paragraph was constructed. We have reorganized and simplified the information and the pertinent information is now the lead in the paragraph. And reads: 

This paper focuses on a subsample of 1027 respondents (34% of the full 2984 sample and 55% of the 1842 usable responses) who answered a set of questions about searching for information to answer a self-nominated clinical question.

4. Results discussing 52-59% stopping search due to satisfaction need to be expanded to also include levels of dissatisfaction as well.

Thanks for pointing out that the implicit dissatisfaction numbers that were included in the original write up were not read as such by you. We have revised the relevant sentence to be more explicit in drawing out the dissatisfaction of search stopping. It reads:

Of the total sample, 52% were satisfied with the answers they found in their first or second attempts. 30% continued on to a third round of search and the rest gave up after their first (10%) or second (9%) attempts due to constraints or dissatisfaction. 

5. The distinction between these 2 statements need to be clarified: "...52% were satisfied with the answers they found in their first or second attempts." and "The percentage of searchers who stopped searching because they found a source that provided enough information in the second attempt (72%) was about the same as in the first attempt (76%)."

Thanks for pointing out that there was needed clarification between the two statements. We have added clauses that should help to make the differences between the two statements clear. The first is a global reporting and the second is a more granular reporting when drilling down in the stages of the search behavior. The statements now read:

Of the total sample, 52% were satisfied with the answers they found in their first or second attempts. 30% continued on to a third round of search and the rest gave up after their first (10%) or second (9%) attempts due to constraints or dissatisfaction. 

 and

When looking at percentages specific to each round of search activity (rather than the overall percentages discussed above), the percentage of searchers who stopped searching because they found a source that provided enough information in the second attempt (72%) was about the same as in the first attempt (76%). 

6. Justify this results statement including the definition of 'worth' being applied: "It was worth trying to find answers via a different route if the first did not work."

Thanks for pointing out that “worth” is a value laden construct. We have removed that inadvertent modifier and re-stated the sentence with more neutral language. The sentence now begins:

Some professionals chose to persist in trying …

7. Again, clarification of the findings is needed similar to item 5 above with the following statement: "Of these, about 34% got the answer they needed. The next most commonly used source in the first round of searching was a general internet search (24%). Of these, 27% stopped searching because the source provided enough information. Those who went to a specific website first (only 8%) had slightly better luck finding information. Of the 8% who used a specific website first, 36% said they stopped searching after the first round because it provided enough information." If each search strategy is reaching less than 40% success, how is overall 76% success found for first search?

Here we were trying to draw out the richness of our data and illustrate the effectiveness of each specific pathway. The percentages mentioned in this paragraph are conditional on the number of individuals who chose that specific strategy. We are at a loss for how to state this information more clearly, and the other reviewers did not seem to have difficulty with this information (or at least did not draw that out). This does not negate your confusion. Our best attempt to address this issue was to add a sentence that states what we are trying do with the information. We added the following statement:

When we drill down into the success of each specific strategy, we see that even the most effective strategy (consulting a friend) was only successful less than half of the time. 

8. Clarify: "general dentists with advanced training were more than twice as likely as general dentists without advanced training to consult a peer-reviewed source first" when these groups' responses were 11% v 13%.

The logit results are not speaking to the numbers of professionals that made a choice (which the percentages do, and may be a relevant interpretation of OLS coefficients), but rather to the odds that a particular search strategy is made given all the other search strategies and characteristics in the model. So, holding all other characteristics constant, a dentist with advanced training is 2 times more likely to choose a peer reviewed source first. We have eliminated the reported percentages to avoid the conflation and confusion of the odds ratio – which is the point of this analysis – not the number of professionals.

9. Sex differences in search stoppage are noted. It should be examined and clarified as to how this may reflect a greater persistence or a lesser effectiveness of searching.

We agree that this is a compelling finding. However, we do not have data to support further examination other than that the differences exist -we cannot answer the “why” here and to do so would go beyond our data. We now note that this finding deserves further investigation in the Discussion. We added the following sentence:

Certainly, the differences found regarding gender identity merit further investigation, and have corollaries in other professional domains (41, 42).

10. As the survey was initiated relative to nicotine cessation and alternative tobacco products, there should be greater presentation of how this specific question was addressed relative to the general behaviors reported.

While the initial survey was relative to novel nicotine products, the specific set of questions under consideration here were not. Instead, they are related to the nominated clinical topic provided from the respondent. Where relevant, we already draw out conclusions specific to novel nicotine. 

11. Figure 2. Remove the number 1 or 2 after reason descriptors as they are distracting to frequency data

Unfortunately, we are unable to remove those numbers, as they are necessary to distinguish between the levels being reported on. Otherwise, the software aggregates the totals.

Reviewer #2: This study tackled a topic that has not been previously explored in the depth and breadth described. This study will encourage more researchers to investigate traits in data retrieval amongst oral health providers to achieve best practice, and more importantly how to address misinformation, and encourage seeking peer reviewed resources.

Thank you for the summary of the paper and your support of the existing version of the work. We think the revisions spurred by the other reviewers have made the manuscript even more crisp. 

Reviewer #3: Great job on drafting an interesting manuscript that addresses an important research question that is warranted in today’s face-paced clinical environment. The manuscript aims to understand the avenues that clinicians take when faced with scenario where they lack information. The authors conducted a survey among more than 3100 clinicians (dentists, hygienists and specialists) to understand the methods clinicians adopt to retrieve information and address uncertainties. Below are my comments:

Background:

Line 102 – I think the sentence needs rewording.

Thank you. We simplified the sentence and adjusted the transition to the following sentence.

Methods:

Line 170 – Why did you choose members of the National Dental Practice Based Research Network as your sampling frame? I know you mentioned later that they were not very different than other clinicians, but I think your audience might need to see a rationale for your choice.

We rephrased the sentence to make it clear the network were our partners in this endeavor (thereby implicitly justifying why we used the network as the sampling frame)

Line 174 – Could you please add the survey instrument as an appendix for the convenience of your audience?

We include a link out to the survey instrument that is archived on the PBRN website.

Line 201 – Dependent variable: What was the rationale behind asking the participants to provide their own case? Was there any way you could have adjusted for the level of complexity of the case the participant came up with? How different do you think that could have been (dentists vs hygienists) (specialists vs GPs)? Do you think it would have been better to have a more homogenous sample (only dentists, only hygienists or only specialists)?

We asked clinicians to nominate a clinical topic because clinicians knowledge bases vary, and what they choose to pursue when presented with uncertainty also varies. So, we could not a priori presume that our sample would pursue a particular question. So we opted to have them nominate their own questions so that we could get at the information important to the study. We added a sentence that helps to ground this decision:

Clinicians have been found to be selective about which clinical issues they pursue based on their perceptions of the tractability of the problem and whether they consider it to be an “important matter” (35). 

Discussion:

I think it’s important to elaborate on the implications of your findings re: clinicians prioritizing time over the credibility of resources from the professional and educational perspectives. What should the dental and dental hygiene associations do to make information more readily available for clinicians? How about integrating some of the researching techniques into school curricula? Do you feel clinicians are equipped with the skills that make their searching strategies efficient?

Thanks for pointing this out. We added a sentence to draw out the merits of looking into these issues in the discussion. Due to space constraints, we were not able to do more than this in the current manuscript. The new sentence reads:

Investigation into how to train professionals on how to do effective searches or how to make high quality sources more readily available may be valuable in reducing the science to service gap in clinical practice

---

## [Decision Letter · Decision Letter 1]

22 Feb 2022

Temporal search persistence, certainty, and source preference in dentistry:

Results from the National Dental PBRN

PONE-D-20-28405R1

Dear Dr. Isett,

We’re pleased to inform you that your manuscript has been judged scientifically suitable for publication and will be formally accepted for publication once it meets all outstanding technical requirements.

Kind regards,

Dragan Pamucar

Academic Editor

PLOS ONE

Additional Editor Comments (optional):

Reviewers' comments:

Reviewer's Responses to Questions

**Comments to the Author**

1. If the authors have adequately addressed your comments raised in a previous round of review and you feel that this manuscript is now acceptable for publication, you may indicate that here to bypass the “Comments to the Author” section, enter your conflict of interest statement in the “Confidential to Editor” section, and submit your "Accept" recommendation.

Reviewer #2: All comments have been addressed

Reviewer #3: All comments have been addressed

2. Is the manuscript technically sound, and do the data support the conclusions?

Reviewer #2: Yes

Reviewer #3: Yes

3. Has the statistical analysis been performed appropriately and rigorously? 

Reviewer #2: Yes

Reviewer #3: Yes

4. Have the authors made all data underlying the findings in their manuscript fully available?

Reviewer #2: Yes

Reviewer #3: Yes

5. Is the manuscript presented in an intelligible fashion and written in standard English?

Reviewer #2: Yes

Reviewer #3: Yes

6. Review Comments to the Author

Reviewer #2: The authors responded to all queries raised by the reviewers, and I have no concerns on dual publication, research ethics or publications.

Reviewer #3: Thank you for your responses to my earlier comments. I have no further comments or concerns. Best of luck!

7. PLOS authors have the option to publish the peer review history of their article (what does this mean?). If published, this will include your full peer review and any attached files.

Reviewer #2: No

Reviewer #3: No

---

## [Editor Report · Acceptance letter]

6 May 2022

PONE-D-20-28405R1 

Temporal search persistence, certainty, and source preference in dentistry:
Results from the National Dental PBRN 

Dear Dr. Isett:

I'm pleased to inform you that your manuscript has been deemed suitable for publication in PLOS ONE. Congratulations! Your manuscript is now with our production department. 

Kind regards, 

on behalf of

Dr. Dragan Pamucar 

Academic Editor

PLOS ONE